# Mechanisms of Demyelination and Remyelination Strategies for Multiple Sclerosis

**DOI:** 10.3390/ijms24076373

**Published:** 2023-03-28

**Authors:** Xinda Zhao, Claire Jacob

**Affiliations:** Institute of Developmental Biology and Neurobiology, Faculty of Biology, Johannes-Gutenberg University, Hanns-Dieter-Hüsch-Weg 15, 55128 Mainz, Germany

**Keywords:** multiple sclerosis, remyelination, immune cells, oligodendrocytes, transcription factors, epigenetics

## Abstract

All currently licensed medications for multiple sclerosis (MS) target the immune system. Albeit promising preclinical results demonstrated disease amelioration and remyelination enhancement via modulating oligodendrocyte lineage cells, most drug candidates showed only modest or no effects in human clinical trials. This might be due to the fact that remyelination is a sophistically orchestrated process that calls for the interplay between oligodendrocyte lineage cells, neurons, central nervous system (CNS) resident innate immune cells, and peripheral immune infiltrates and that this process may somewhat differ in humans and rodent models used in research. To ensure successful remyelination, the recruitment and activation/repression of each cell type should be regulated in a highly organized spatio–temporal manner. As a result, drug candidates targeting one single pathway or a single cell population have difficulty restoring the optimal microenvironment at lesion sites for remyelination. Therefore, when exploring new drug candidates for MS, it is instrumental to consider not only the effects on all CNS cell populations but also the optimal time of administration during disease progression. In this review, we describe the dysregulated mechanisms in each relevant cell type and the disruption of their coordination as causes of remyelination failure, providing an overview of the complex cell interplay in CNS lesion sites.

## 1. Introduction

Multiple sclerosis (MS) is an inflammatory autoimmune disease and the most frequent degenerative disease of the central nervous system (CNS) [1]. Immune cells with peripheral origin pass through the damaged blood–brain barrier (BBB) and release cytokines, including tumor necrosis factor alpha (TNFα), interferon gamma (IFN-γ), and interleukin 17 (Il-17). These cytokines can directly attack myelinating oligodendrocytes (OLs) [2] or indirectly impair the OL–neuron coupling by polarizing microglia into the M1 (pro-inflammatory) state, which then activate reactive neurotoxic A1 astrocytes by secreting Il-1α, TNFα, and the complement component 1q (C1q) [3]. The combined secretion of cytokines by lymphocyte infiltrates, activated M1 microglia, and A1 astrocytes lead to OL death and demyelination [4,5]. Demyelinated axons lose insulation and metabolite support from OLs [6,7], which eventually leads to axonal degeneration and neuronal loss [8].

Although remyelination can be demonstrated by the occurrence of thin myelin using electron microscopy and animal models at different time points after a demyelinating lesion [9], it has been difficult to conclude whether the thinly myelinated axons in postmortem MS patients are partly demyelinated or incompletely remyelinated axons [10]. However, Bodini et al. used positron emission tomography to show in vivo myelin degeneration and repair by tracing myelin-binding Pittsburgh compound B [11]. This provided evidence that remyelination can occur in some MS lesions. Imaging remyelination in living MS patients remains, however, challenging, and currently, the most accurate way to evaluate remyelination is to measure visual evoked potential (VEP) latency (reviewed in [12]), which requires a pre-existing lesion in the optic nerve. Remyelination is mostly seen in acute and relapsing/remitting MS (RRMS) active lesions and borders [13]. Mixed/inactive lesions, marked by decreased or even absent remyelination, present a rim enriched in microglia and pro-inflammatory iNOS+ myeloid cells and depleted in anti-inflammatory CD163+ myeloid cells, compared to active lesions [13]. Furthermore, an increased density of CD3+ T lymphocytes together with phagocytic and activated microglia has also been shown in normal-appearing white matter (NAWM) of MS brain tissue compared to control WM [13,14]. The alteration of innate immune cell populations in MS patients varies largely within the lesion areas and throughout disease progression.

The density of OL progenitor cells (OPCs), the main cell population remyelinating axons, increases in early MS lesions compared to the adjacent NAWM, which sustains mature OL populations with unaltered density. Chronic MS lesions are marked by severe depletion of OPCs and mature OLs [15]. Consistently, chronic MS lesions were found to have not only a decreased OL density but also a decreased proportion of OLs that expressed myelin gene regulatory factor (MYRF) compared to NAWM [16].

This heterogeneity of cell populations and associated extent of remyelination provide insights into the detrimental and beneficial roles of each cell type in MS progression and repair at different spatio-temporal regions.

## 2. Roles of Peripheral and CNS Resident Innate Immune Cells

Demyelination results in large amounts of myelin debris, composed of cholesterol, phospholipids, glycolipids, and myelin-associated proteins [17]. These myelin debris are not only neurotoxic [18] but also inhibit axonal outgrowth [19,20] and remyelination [21]. Moreover, myelin-associated proteins, such as myelin basic protein (MBP), myelin proteolipid protein (PLP), and myelin oligodendrocyte glycoprotein (MOG), serve as antigens to induce an adaptive autoimmune response in MS patients [22,23].

Clearance of myelin debris by macrophages and microglia following demyelination is a crucial process to allow remyelination. Phosphatidylserine is a phospholipid that is abundant in myelin. During OL apoptosis, phosphatidylserine translocates to the outer layer of the plasma membrane, where it serves as a signal inducing phagocytosis [24]. Indeed, phosphatidylserine mediates the activation of TAM (Tyro3, Axl, and Mertk) receptors through their ligands Protein S and Gas6 [25,26]. Consistently, in a cuprizone model of CNS demyelination and remyelination, Shen et al. found that *Mertk*-KO mice had impaired myelin clearance and remyelination [27]. Another important phospholipid sensor is the triggering receptor expressed on myeloid cells-2 (TREM2), which is also expressed by microglia and macrophages [28]. TREM2 is essential for microglia activation and phagocytosis. *Trem2*-deficient mice were found to have fewer lipid droplets, elevated endoplasmic reticulum (ER) stress, and enhanced neuronal damage and motor impairments [29,30]. On the other hand, treatment with the TREM2 monoclonal agonistic antibody ALoo2a enhanced myelin debris clearance in the mouse model of demyelination mediated by cuprizone [28]. The same study also detected elevated TREM2 expression in lipid-laden macrophages–microglia in active lesions from MS patients but not in NAWM and control WM; this presents further evidence that the activation of microglia and myelin clearance also occurs transiently in active lesions.

Pattern recognition receptors (PRRs) expressed in CNS phagocytes can also sense myelin debris. Major PRRs involved in CNS myelin clearance include Toll-like receptors (TLRs), Nod-like receptors (NLRs), and C-type lectin receptors (CLRs) [31]. Deerhake et al. showed that PRRs have both pathologic and protective roles in the experimental autoimmune encephalomyelitis (EAE) model, which is commonly used to model the autoimmune inflammatory aspect of MS [31]. Indeed, after deleting the primary TLR adaptor protein Myd88, mutant mice had fewer CD4+ infiltrates and partial-to-complete resistance to MOG immunization [32,33]. However, using the lysolecithin model of CNS demyelination, Cunha and colleagues showed that *Myd88* mutant mice had suppressed phagocyte activation and impaired myelin clearance and remyelination [34]. These seemingly contradictory phenotypes are partially due to the fact that the former studies used the EAE model, whereas the latter chose the lysolecithin model. As lysolecithin directly dissolves myelin and kills OLs at lesion sites without causing a major immune response from the peripheral immune system, it is arguably more suitable for CNS myelin clearance and remyelination studies.

During demyelination, the large amount of cholesterol present in myelin debris is phagocytosed by activated microglia and macrophages [35]. Activated phagocytes upregulate sterol synthesis genes in mice [35] and in MS repairing lesion samples [36], except for the gene coding for 24-dehydrocholesterol reductase (DHCR24). DHCR24 leads to the sterol synthesis intermediate product desmosterol that activates LXR (liver X receptor) signaling in phagocytes, which was shown to resolve inflammation and enhance recycled cholesterol export from phagocytes for remyelination by OLs in vivo [35]. In aged phagocytes, cholesterol excessively accumulates, switching from a free to a crystal form, which activates inflammasomes [37]. 

On the other hand, activated microglia release the cytokines Il-1α and TNF, as well as C1q, which polarizes astrocytes into a reactive neurotoxic A1 state. Unlike M1 microglia which actively engulf myelin debris, A1 astrocytes display decreased *Mertk* expression and loss of phagocytic capacity of myelin debris [3]. Furthermore, A1 astrocyte-conditioned medium can induce neuron and OL apoptosis in vitro [3]. Although activated phagocytes are recruited to lesion sites and clear myelin debris, persistent microglia and astrocyte activation are associated with chronic inflammation and loss of myelin. As mentioned earlier, histological analyses showed increased levels of TMEM119+ homeostatic microglia and iNOS+ (pro-inflammatory) myeloid cells and decreased levels of CD163+ (anti-inflammatory) myeloid cells in the periphery of mixed lesions compared to active lesions [13]. Mixed lesions are also characterized by fewer oligodendrocytes and less myelin in the lesion center compared to active lesions that can be remyelinated and to NAWM [13]. 

## 3. Roles of Oligodendrocyte Lineage Cells in Remyelination

### 3.1. Oligodendrocyte Progenitor Cells Migrate to Lesion Sites and Proliferate Following CNS Injury

Both parenchymal adult OPCs and subventricular neural stem cells (NSCs) can repopulate depleted OLs in lesion sites [38,39,40,41,42,43,44]. The relative contribution of OPCs versus NSCs depends on the location of lesions. Using cuprizone-conditioned mice, Xing and colleagues found that in the rostral region adjacent to the subventricular zone, the majority of OLs originated from Nestin+ neural precursor cells (NPCs) and formed remyelinated internodes with a thickness equivalent to unchallenged controls, in contrast to the remyelinating sheath observed in other CNS regions that is typically thinner than in the unlesioned CNS [43]. In terms of distribution, however, the NG2 (nerve/glial antigen-2)- and PDGFRα (platelet-derived growth factor receptor alpha)-expressing OPCs are more widespread throughout the CNS than NPCs and might, hence, likely be more available for OL repopulation for the various CNS lesion sites [38,39,42,45,46]. OL repopulation and subsequent remyelination are relatively efficient in active demyelinating lesions, even after several successive lesions, but the efficiency of this process decreases in chronic MS lesions [13,15]. Demyelination induces the release of a range of chemoattractants that activate OPC migration to lesion sites [45,47,48]. Moyon and colleagues identified the chemoattractants *Il1β* and *Ccl2*, among other upregulated genes, in OPCs isolated from cuprizone-treated mice compared to healthy untreated mice [45]. CCL2 was also found to be upregulated at the protein level in active MS lesions and cuprizone-treated mice. Consistent with their chemoattractant function, in vitro treatment with IL1β or CCL2 promotes OPC migration [45]. In addition to chemoattraction, other mechanisms are involved in OPC migration. For instance, the PDGF-A-induced ERK pathway and the interaction of integrins in OPCs with laminin in the extracellular matrix (ECM) promote OPC migration and process extension via focal adhesion kinase activation and actin reorganization [48,49].

On the other hand, a wide range of chemorepellents is released by CNS cells during CNS demyelination [50]. Chondroitin sulfate proteoglycans (CSPGs) inhibit remyelination through binding to their cognate receptor, protein tyrosine phosphatase σ (PTPσ), on OPCs [51]. Inhibiting CSPG/PTPσ signaling leads to increased expression of matrix metalloproteinase-2 (MMP-2) in OPCs, digestion of inhibitory CSPGs by MMP-2, and enhanced recovery from EAE induction [51]. A review by De Jong et al. [50] provides a detailed overview of the complex ECM remodeling process during demyelination. Improper ECM remodeling may inhibit OPC migration and OL repopulation and lead to remyelination failure in MS lesions. For example, Boyd and colleagues (2013) reported that the chemoattractant rSema3F appears in and around active lesions, together with astrocytes, microglia, and macrophages, whereas Sema3A is absent [47]. Sema3A appears only in chronic active MS lesions and inhibits mouse OPC migration in vivo. All in all, upon CNS injury, CNS resident cells secrete a wide range of chemicals, ECM, and ECM-remodeling enzymes, providing molecular and structural cues for OPC migration. In pathological conditions, ECM chemorepellents can exceed chemoattractants, hence inhibiting OPC migration to lesion sites, leading to inefficient repair. 

OPC migration to lesion sites precedes proliferation [52]. A wide range of ECM molecules, growth factors, and chemokines secreted by astrocytes are crucial for OPC proliferation. Endothelin-1 (ET-1) inhibits OPC maturation, thereby maintaining their migratory and proliferative capacity [53]. ET-1 also increases the production of the growth factors PDGFA and FGF2 and promotes OPC proliferation via activating ERK/MAPK and CREB pathways [54]. Various ECM components, such as fibronectins and laminins, appear or are upregulated in active MS lesion sites. They can self-polymerize or assemble to form an adhesive bridge between OPCs and the surrounding tissue via integrin receptors, promoting OPC migration and proliferation [50,55,56]. 

Not surprisingly, some factors that transiently promote migration and proliferation such as Wnt also inhibit OPC differentiation [57,58,59], which protects premature OLs from insults within the inflamed microenvironment [60]. Wnt signaling was shown to be activated after demyelination, and this activation is associated with the increased expression of its intranuclear mediator Tcf4 in mouse lesion areas and MS lesions [57]. Wnt signaling leads to the activation and accumulation of β-catenin, which eventually translocates to the nucleus and triggers expression of the Cxcl12-binding chemokine receptor Cxcr4 that facilitates OPC migration [59]. Perturbed Wnt/β-catenin signaling may compromise remyelination. The overactivation of β-catenin under the control of the *Olig2* promoter in mice inhibits OPC differentiation and delays remyelination after lesion [57]. Conversely, overexpression of the Wnt inhibitor Apcdd1 increases OPC differentiation in vitro and enhances remyelination after lysolecithin lesion [61]. 

Similarly, the activation of the Notch pathway allows OPC proliferation [62], inhibits their differentiation [53,63], and may, therefore, be a target for remyelination [64]. The Notch pathway is activated by the membrane-bound ligands delta or serrate (Jagged) secreted by reactive astrocytes [60]. Expression of the Notch ligands Jagged 1 and Jagged 2 increases at the lesion margins following CNS demyelination [65]. The activation of Notch in OPCs enhances their proliferation in demyelinated lesions and prevents their differentiation [60]. Notch signaling can modulate the activity of Sox10, a major transcription factor for myelination and remyelination, via the Notch downstream effector Hes5, which sequesters Sox10 by direct binding [66]. On the other hand, the inhibition of Notch1 restricts OPC expansion and induces differentiation and myelination [64]. Indeed, the ablation of Notch1/2 under the control of the *Olig1* promoter results in accelerated remyelination following lysolecithin demyelination compared to control littermates, albeit at the expense of OPC proliferation [64]. However, Notch also affects the fate of neurons and other glial cells in the CNS [60]. Therefore, one must consider the off-target effects when developing MS therapies involving the modulation of the Notch pathway. 

### 3.2. Key Transcription Factors for Oligodendrocyte Maturation and (Re)myelination

Olig2 is expressed in the entire OL lineage cells and is an essential transcription factor for oligodendrogenesis [67]. The loss of Olig2 in NG2 cells leads to reduced OPC production [68] and to a fate switch into the astrocyte lineage [69]. The overexpression of Olig2 under control of the *Sox10* promoter in mice enhances the expression of *Myrf*, *Mbp,* and *Plp1* expression, promoting OPC differentiation and enhancing remyelination after lysolecithin lesion [70]. Olig2 expression remains relatively low in healthy WM while increasing in active but not inactive MS lesions, which suggests its contribution to the success of remyelination [70]. Olig1, a homolog to Olig2, is co-expressed with Olig2 in most OL lineage cells. Unlike *Olig2*-null mice that die at birth, *Olig1*-null mice exhibit a normal phenotype until the adult stage. Additionally, Olig2 is confined to the nuclear compartment, whereas Olig1 is mostly localized in the cytoplasm of OLs. During demyelination and remyelination, however, Olig1 translocates to the OL nucleus. This subcellular translocation is very likely to be critical for remyelination, as suggested by the finding that *Olig1*-null mice fail to express PLP and MBP and to remyelinate after cuprizone- or lysolecithin-induced CNS demyelination, while control littermates remyelinate extensively [71].

Nkx2.2 is transiently expressed at the onset of OPC differentiation and determines the differentiation timing. Zhu and colleagues have reported that Nkx2.2 represses PDGF signaling via the downregulation of PDGFRα, arresting OPC migration and proliferation while inducing their differentiation in the developing mouse spinal cord [72]. From a therapeutical point of view, it could be interesting to target Nkx2.2 in MS lesions; however, it may also be challenging due to the transient time window of Nkx2.2 expression. Hypothetically, activating Nkx2.2 in MS lesions while OPCs migrate to or proliferate at lesion sites could result in reduced numbers of OPCs that can differentiate into remyelinating OLs. Similarly, one should also consider the spatio-temporal dual function of the PDGF signaling, as opposed to merely pro-proliferative or inhibitory to differentiation. 

Sox10 is required during OPC differentiation and induces myelination-associated gene expression (reviewed in [73]). During postnatal development in mice, Sox10 deletion in OLs causes hypomyelination associated with a drastically decreased expression of mature OL markers, such as *Plp*, *Mbp*, and *Myrf* [74]. In addition, Myrf was completely absent after additional ablation of Sox8, a close relative of Sox10 [74]. As Sox10 upregulates myelination-associated genes during OL developmental myelination, it is tempting to speculate on its role in remyelination. Indeed, Duman and colleagues reported that an increased expression of the chromatin-remodeling enzyme histone deacetylase 2 (HDAC2) enhanced MBP expression and CNS remyelination after lysolecithin lesion in mice, through Sox10 stabilization and maintenance of Sox10 target genes activation [75]. 

The role of extracellular signal-regulated kinases (ERKs) in OPC differentiation is less clear. One study found that *Erk2*-null GFAP-expressing radial glial cells, which give rise to neurons and oligodendrocytes, failed to differentiate from OPCs to mature OLs in vitro and that postnatal OL differentiation and myelination were delayed in the mouse corpus callosum, suggesting a role of ERK2 in the timing of OPC differentiation and myelination [76]. However, other studies did not find evidence linking ERK1/2 activity to OPC differentiation [77,78,79]. Instead, the latter studies show that ERK1/2 signaling is mostly required for myelin thickness. *Erk1/2*-null mice showed significant hypomyelination, while the size of the PLP+ OL population was similar to that of control littermates [77]. In comparison, mice under sustained ERK1/2 activation displayed thicker remyelination of spinal cord lesions 7 weeks after lysolecithin-induced demyelination than control littermates [79]. Consistently, the FDA-approved medication miconazole has been shown to activate ERK1/2 specifically in OPCs and to enhance remyelination after lysolecithin lesion in mice [80]. Interestingly, ERK activation in preexisting OLs promotes the formation of new myelin sheaths [81], which challenges the view—also challenged later by Duncan et al. [82]—that remyelination is conducted only by OPCs.

OLs require Myrf for myelin formation [83], maintenance [84], and remyelination [16]. Mice with *Myrf* ablated in the OL lineage under the control of the *Olig2* promoter show decreased myelin gene expression compared to control littermates and fail to myelinate [83]. Conversely, overexpression of Myrf induces MBP expression in developing chick spinal cord [83]. Conditional *Myrf* ablation in mature OLs in adult mice leads to the downregulation of myelin genes, including *Plp*, *Mbp*, *Mog*, and *Mag*, without, however, affecting Sox10 expression [84]. Indeed, Sox10 activates the expression of *Myrf* at the transcriptional level [73] and in turn Myrf guides Sox10 target gene selection during OL differentiation [85] and cooperates with Sox10 to activate the transcription of myelin genes [74]. However, Sox10 expression is not regulated by Myrf [84]. Consistently, the depletion of *Myrf* in OPCs under the control of the *Pdgfra* promoter in mice leads to myelin gene expression and remyelination failure after lysolecithin-induced demyelinating lesion in the CNS, while OPC recruitment to the lesion site remains unaffected [16]. Furthermore, the same study reported that fewer Sox10+/NogoA+ cells were Myrf+ in chronic MS lesion centers than in shadow plaques, indicating that a lack of Myrf may contribute to remyelination failure in some MS lesions. 

The tyrosine kinase Fyn promotes OPC differentiation, OL process extension, and myelination [86]. *Mbp* mRNA-containing granules can be shuttled to OL–axon contact sites, where L1/contactin-activated Fyn phosphorylates heterogeneous nuclear ribonucleoprotein A2 (hnRNP A2), leading to the dissociation of the RNA transport granules, thereby allowing for spatio-temporal regulation of MBP translation and myelination [87]. Additionally, upon activation by the laminin family member Netrin-1, Fyn can inactivate Rho-A, which is a downstream effector of LINGO-1, a negative regulator of OPC differentiation [88,89]. Another laminin family member, laminin-2, which interacts with β1-integrin, can also initiate myelination via Fyn activation. Consistently, the ablation of laminin-2 in the mouse CNS leads to delayed OL maturation and hypomyelination in vivo [90].

### 3.3. Epigenetic Modulation of Myelination in Oligodendrocyte Lineage Cells and the Aging Process

HDACs are known to repress gene expression by deacetylating histones, which leads to chromatin condensation and thereby limits access to genes for the transcriptional machinery. In addition, HDACs have many non-histone targets, such as transcription factors and other factors involved in transcriptional regulation. Class 1 HDACs are powerful regulators of OPC differentiation, myelination, and remyelination [75,91,92,93]. HDAC2 prevents the targeting of Sox10 to the proteasome via deacetylating its negative regulator eukaryotic elongation factor 1A1 (eEF1A1) [75] and thereby promotes Sox10-mediated activation of promyelinating and myelin genes, such as *Myrf* and *Myelin basic protein* (*Mbp*) in OLs [94]. Theophylline, a potent HDAC2 activator when used at a low dose, increases Sox10 and myelin protein expression and remyelination in the mouse spinal cord after a demyelinating lesion by lysolecithin in young and old adults [75], in the mouse sciatic nerve after nerve crush injury [75] and in a mouse model of peripheral neuropathy [95]. On the other hand, co-immunoprecipitation analyses revealed that the association of HDAC1/2 with the transcription factor Yin Yang 1 (YY1) was weak in OPCs but enhanced in OLs [93]. In this study, the authors show that YY1 inhibits the expression of *Tcf4* and *Id4* by recruiting HDAC1 to their promoter region. Similarly, HDAC1 was found to be increasingly recruited to the promoter of the differentiation inhibitor *Hes5* in the mouse corpus callosum after demyelination induced by cuprizone treatment, and this was associated with an increased expression of *Olig1* [96]. Taken together, these studies show that class 1 HDACs can enhance the expression of multiple promyelinating factor and myelin genes by repressing their inhibitors. 

Insufficient OPC differentiation into OLs contributes to the failure of remyelination in MS patients. Although EAE animal models showed that mature OLs can be completely repopulated, even after four episodes of induced cortical demyelination, postmortem brains of chronic MS patients showed decreased numbers of OPCs and OLs in cortical lesions [15]. In this study, EAE rats were, however, between 8 and 9 months old when sacrificed, whereas the postmortem MS tissue came from patients who were on average 54 years old and had suffered from chronic MS for decades [15]. One possible explanation for the decrease in OL repopulation in chronic MS patients is aging [97,98]. Interestingly, a recent phase 2a study showed that bexarotene, a retinoic acid receptor RXR-gamma agonist with CNS pro-remyelinating effects demonstrated in preclinical studies [99], improved VEP latency in patients with chronic optic neuropathy aged up to early 40s but not older [100]. Similarly, remyelination is also less efficient in aged animals after lysolecithin- or cuprizone-mediated demyelination [96,101]. Indeed, RNA sequencing showed evidence of age-dependent epigenetic control of OPCs. Approximately 20% of all genes are differentially expressed in aged OPCs compared to young OPCs [102]. Particularly, young OPCs show higher expression of genes related to self-renewal, such as *Pdgfra*, *Ascl1*, and *Ptprz1*, whereas aged OPCs express higher levels of the early differentiation markers *Cnp1*, *Sirt2*, and *Enpp6*, indicating a loss of the stem cell characteristics that are essential in OPC proliferation and differentiation [102]. Furthermore, Shen et al. showed that after cuprizone treatment, *Cnp*, *Mag*, *Olig1*, and *Hdac* transcripts were upregulated in young but not old mice [96]. In addition, old mice displayed decreased numbers of HDAC1- and HDAC8-expressing OPCs and decreased remyelination compared to young mice [96]. Interestingly, old mice also showed a higher percentage of OPCs expressing Sox2 – a transcription factor that maintains multipotency and, hence, inhibits differentiation – than young mice [96]. The authors, therefore, proposed that inefficient class 1 HDAC activity in old mice may lead to the misregulation of transcription factors for OPC differentiation, resulting in reduced remyelination. 

Sirtuin 2 (SIRT2), a member of class III HDACs, whose activity depends on nicotinamide adenine dinucleotide (NAD+), has also been implicated in the decline of OPC remyelination capacity during aging. Indeed, Ma and colleagues found that nuclear localization of SIRT2 was impaired in the OPCs of aged mice [103]. In this study, the authors show that SIRT2 expression is high and largely nuclear in OPCs during postnatal myelination in mice but decreases in adults and is only detected in the cytoplasmic compartment of mature OLs. After a demyelinating lesion by lysolecithin, SIRT2 is re-expressed in most OPCs and largely localized in the OPC nuclear compartment in young adult mice, whereas re-expression and nuclear localization are a lot lower in aged mice. Remarkably, supplementing β-nicotinamide mononucleotide, the precursor of NAD+, rescues SIRT2 re-expression and nuclear localization and remyelination in aged mice [103]. 

Reversing age-dependent defects intrinsic to OPCs proved to enhance OPC differentiation and remyelination [102]. In this study, metformin, a drug already used for the treatment of type 2 diabetes, was found to improve the mitochondria function of aged OPCs via modulating the AMPK pathway, increased the expression of *Pdgfra* and *Ascl1* in OPCs, and enhanced remyelination after a CNS lesion induced by ethidium bromide. The authors suggested that the pro-remyelinating effect of metformin may be due to the fact of improved DNA repair and increased autophagy, which are known effects of metformin. Indeed, another study found that metformin increased autophagy activity in microglia and promoted the clearance of myelin debris in rat spinal cord lesions [104].

In addition to aging, Segel and colleagues found that the ECM present in the OPC microenvironment stiffens over time and that expression of the “stiffness” mechanoresponsive ion channel PIEZO1 also increases in OPCs during aging [105]. In this study, the authors show that transplanted neonatal *Piezo1*-null OPCs but not transplanted control OPCs keep proliferating in the aged rat cortex. This work indicates that PIEZO1 acts as a key mediator of OPC mechanical signaling and that ablation of PIEZO1 disables OPC response to the stiffened microenvironment and thereby allows OPCs to maintain their activity during aging.

The different cell types, cell processes, and molecular players involved in CNS demyelination and remyelination described above are illustrated in Figure 1.

Peripheral immune infiltrates and CNS resident cells secrete pro-inflammatory cytokines and neurotoxic substances leading to oligodendrocyte (OL) death and demyelination. Remyelination by OLs requires a choreographed network of both extrinsic and intrinsic factors that promote OL repopulation, differentiation, and remyelination. OPC: OL precursor cell; NSC: neural stem cell; M1: M1 microglia; M2: M2 microglia; A1: A1 astrocyte; A2: A2 astrocyte; dOL: dying OL. The figure was generated using Biorender.com.

## 4. Treatment Perspectives for Remyelination

### 4.1. Agents That Directly Promote OPC Differentiation and Remyelination

Various high-throughput screening methods provide insights into candidate molecules that promote OPC differentiation [106,107]. Using OPCs cultured in micropillar plates, Mei and colleagues identified eight FDA-approved anti-muscarinic drugs, namely, atropine, ipratropium, oxybutynin, trospium, tiotropium, quetiapine, benztropine, and clemastine, that enhance OPC differentiation in vitro [107]. Clemastine was also verified to promote OPC differentiation and remyelination in lysolecithin-challenged mice [107] and in EAE mice [108]. Consistently, clinical trials in MS patients showed a small but significant reduction in VEP latency in patients treated with clemastine, indicating remyelination [109], and lower levels of plasma neurofilament (NF) light-chain in MS patient blood samples [110], which is used as an indirect marker of neuroprotection/neurodegeneration. With multiple candidate agents proven to be ineffective in phase 3 clinical trials (see Table 1), clemastine is currently the only agent under phase 3 clinical trials for remyelination.

A variety of additional therapeutic compounds that promote OL maturation and remyelination via distinct pathways have been discovered. Some of the targeted pathways and transcription factors have already been discussed. For example, benztropine downregulates the Notch1 pathway and promotes rodent OPC differentiation and myelination in vivo [106]. Theophylline increases HDAC2 expression and activity, which increases Sox10 expression in mouse OLs and enhances remyelination in vivo [75]. Miconazole increases ERK1/2 phosphorylation of mouse OPCs in vitro and promotes remyelination in lysolecithin-induced mouse lesion models [80]. The promising preclinical evidence, supported with pathways that are well-known for OPC differentiation or myelination, warrants interest to initiate clinical trials. On the other hand, opicinumab, an antibody that blocks the myelination-negative regulator LINGO-1 (leucine-rich repeat and Ig domain-containing NOGO receptor interacting protein-1) was found to improve remyelination in vivo [113] and was well tolerated in a phase 1 clinical trial [114] but did not show any significant improvement in remyelination of MS lesions in a phase 2 clinical trial [115]. This is the first compound that has been tested for myelin repair in MS patients. Since then, a few other candidates were concluded to not affect remyelination in MS patients: biotin, at high dose, can promote myelin synthesis via the cofactors ACC1/2 (acetyl-CoA carboxylase) and was shown to improve motor and visual function in a phase 1 trial [111] but showed no significant improvement in walking in a phase 3 trial [112]. Domperidone was shown to increase serum prolactin in humans [118], which improves remyelination in animal models [125]. Albeit increased serum prolactin levels were found in MS patients given domperidone, the treatment showed no evidence of an effect on disease progression in a phase 2 clinical trial [119]. The administration of olesoxime in mouse demyelination models improved OPC maturation and remyelination [123] and was tested to be safe and well tolerated in a phase 1 clinical trial [124]. As mentioned, bexarotene activates RXR-gamma and thereby stimulates OPC differentiation. Interestingly, its effect on VEP improvement was only significant in patients under 43 years old, which indicates an age-related decline in remyelination capacity [99,100].

### 4.2. Rejuvenating Aged OPCs to Repopulate Remyelinating OLs

The hypothesis that aged OPCs have impaired differentiation capacity makes metformin an attractive candidate to promote remyelination in MS. As already mentioned above, Neumann and colleagues showed that metformin improves remyelination in aged animals, most likely by increasing autophagy and reducing DNA damage, which may contribute to remyelination [102]. This agent is currently being tested in several clinical trials for MS patients, including a phase 2 trial where metformin and clemastine are administered in combination (NCT05349474, NCT04121468, NCT05131828, and NCT05298670).

### 4.3. Preventing OPC Differentiation Arrest and OL Death Mediated by an Inflamed Environment

As discussed, the elevated neurotoxic cytokines and their induced oxidative stress present in MS lesions, especially with prolonged persistence, can induce OL death and inhibit OPC differentiation. Therefore, compounds with antioxidative and anti-apoptotic properties may be potent candidates to ameliorate MS progression and to improve remyelination. The compound rHIgM22 was found to co-localize with integrin β3 and to upregulate its downstream Src family kinase (SFK) Lyn in rat OLs, preventing OL apoptosis [121]. This study shows that a single dose of rHIgM22 infusion following an MS relapse is well tolerated and can be detected in patients’ cerebrospinal fluid in a dose-dependent manner, which warrants further studies [122]. The antipsychotic drug, quetiapine, is found to block the cuprizone-induced elevation of lipid peroxides via antioxidative function. Lipid peroxides inhibit OPC differentiation. Adding quetiapine in cuprizone-treated rat primary OPCs promotes OPC differentiation in vitro [120]. The phase 1 clinical trial on quetiapine (NCT02087631) was completed but not published. Another candidate agent with antioxidative and anti-apoptotic properties, erythropoietin, was shown to improve the motor function of a small group of MS patients in a phase 1 clinical trial but showed no clinical significance in the subsequent phase 2 trial [116,117].

## 5. Concluding Remarks

As discussed, chronic lesions are associated with prolonged inflammation, disturbed signaling pathways, and altered epigenetic control within the CNS, all of which contribute to remyelination failure. Proper oligodendrogenesis and remyelination depend on both extrinsic and intrinsic factors, and these factors should be modulated appropriately at different stages of disease progression. For example, repressing microglia activity at an early stage of the disease can result in inefficient myelin clearance, which impedes re-myelination [34], whereas inactivation of Notch signaling in OL lineage cells enhances their differentiation but compromises their proliferation. Therefore, effective strategies for treating MS should implement modulatory or synergistic treatments at different stages of MS pathology or at certain time points after a demyelinating event. From the early stages of the disease onward, it is critical to focus on repressing the overactivation of the innate and adaptive immune system to preserve mature OLs and their myelin. Following a demyelinating event, enhancing CNS phagocytic activity to remove myelin debris would be important to provide an optimal microenvironment for subsequent remyelination. Pro-migratory or pro-proliferative strategies may be difficult to combine with pro-differentiating and pro-myelinating interventions. Therefore, ideally, OPC differentiation should be promoted when sufficient numbers of OPCs have already migrated into the lesion sites. A lot of research is, however, still needed to accurately pinpoint the time window where OPC differentiation is preferably induced in MS patients, and this is very likely dependent on each patient. To achieve this goal, it appears essential to identify biomarkers of disease stage, demyelination events, and axonal loss that are easily accessible, for example, in the blood or other body fluids, such as plasma levels of NF light-chain that can be used as an indirect marker of axonal degeneration. Such markers would be very instrumental to decide on the best treatment strategy in a given MS patient at a given time.

## Figures and Tables

**Figure 1 ijms-24-06373-f001:**
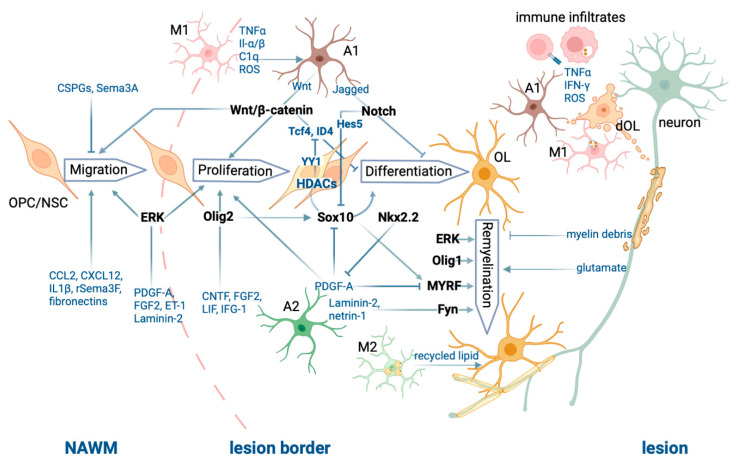
Summary of the processes of CNS demyelination and remyelination.

**Table 1 ijms-24-06373-t001:** Preclinical results and clinical trial outcomes related to CNS remyelination.

Compound	Mechanism of Action—Preclinical	Phase 1	Phase 2	Phase 3
Clemastine	Antimuscarinic [107]	VEP100 latency reduced in treated group [109]	Plasma NF light-chain lower in treated group [110]	Recruiting (NCT05338450)
Biotin	Improves fatty acid synthesis via cofactors ACC1/2 [111]	Improved motor and visual function [111]		No significant improvement in walking [112]
Bexarotene	RXR-gamma agonist [99]		Improved VEP latency only in patients up to 43 years old [100]	
Opicinumab	Blocks LINGO-1 signaling [113]	Safe and well tolerated [114]	No significant remyelination [115]	
Erythopoietin	Prevents brain atrophy [116]	Improved motor function in treated group [116]	No clinical significance [117]	
Domperidone	Promotes prolactin secretion [118]		Did not slow disease progression [119]	
Quetiapine	Antioxidative and pro-proliferative to OPCs [120]	Completed but not published (NCT02087631)		
rHIgM22	Inhibits apoptosis via Lyn kinase [121]	Safe and well tolerated [122]		
Benztropine	Downregulates Notch1 [106]			
Metformin	Rejuvenates OPCs and improves autophagy [102]	Recruiting (NCT05349474, NCT04121468, NCT05131828, NCT05298670)		
Theophylline	Activates HDAC2, increases Sox10 [75]			
Miconazole	Phosphorylates ERK1/2 [80]			
Olesoxime	Directly stimulates OPC maturation [123]	Safe as an add-on therapy to interferon-beta [124]		

## Data Availability

Not applicable.

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
