# Peer review of "Mechanisms of Demyelination and Remyelination Strategies for Multiple Sclerosis"

_ijms, 2023, doi:10.3390/ijms24076373_

Round 1

Reviewer 1 Report

An extensive original review about the cellular mechanisms implicated in multiple sclerosis. The authors analyze the pattern of regulation and dysregulation of multiples cells type related to remyelination. The study design is appropriate and the analysis is well argued.    

Author Response

We are very thankful to Reviewer 1 for the positive comments about our manuscript.

Reviewer 2 Report

The present review article by Zhao and Jacob entitled “Mechanisms of demyelination and remyelination strategies for multiple sclerosis” discuss the importance of fine coordination between demyelination/remyelination with specific relevance to multiple sclerosis. Authors also reviewed the recent developments in dissecting the cell type specific (oligodendrocyte lineages) mechanisms in multiple sclerosis that could provide drug candidates targeting the optimal microenvironment at lesion sites for remyelination. Authors have presented very systematic literature covering all relevant research developments in the field. Schematic figure and table nicely represent a brief summary connecting the role and significance of demyelination and remyelination strategies for multiple sclerosis.

Author Response

We are very thankful to Reviewer 2 for the positive comments about our manuscript. We have corrected the English language throughout the text by using Grammarly. We think the manuscript is improved now.

Reviewer 3 Report

The manuscript entitled " Mechanisms of demyelination and remyelination strategies for 2 multiple sclerosis" is a well-written paper. The topic is well chosen and has the potential to compile and present current developments to interested researchers. The first chapter of the manuscript on MS pathology and myelination is well-structured and supported by current literature. However, the last chapter on MS therapeutic strategy, the most crucial part of MC, was very superficial. This section needs to be expanded further within the knowledge of the literature. As such, the MC remains far from achieving the initially promised goal. Therefore, this section should be recreated with sub-headings. Not only the molecules used in clinical studies but also the molecules mentioned in basic research with potential remyelination potential can also be included.

Author Response

We thank Reviewer 3 for the positive comments about our manuscript. As requested, we have expanded the last chapter about potential remyelinating treatments and we have subdivided this chapter into sub-chapters. We think this section is more substantial now and the manuscript improved.

Round 2

Reviewer 3 Report

The suggested corrections have been tried to be made. The MC is acceptable as it is.